# Implementing an Antimicrobial Stewardship Programme to Improve Adherence to a Perioperative Prophylaxis Guideline

**DOI:** 10.3390/healthcare10030464

**Published:** 2022-03-01

**Authors:** Nehad J. Ahmed, Ziyad S. Almalki, Abdullah A. Alfaifi, Ahmed M. Alshehri, Abdullah K. Alahmari, Emad Elazab, Alaa Almansour, Abdul Haseeb, Mohamed F. Balaha, Amer H. Khan

**Affiliations:** 1Clinical Pharmacy Department, College of Pharmacy, Prince Sattam Bin Abdulaziz University, Al-Kharj 16273, Saudi Arabia; n.ahmed@psau.edu.sa (N.J.A.); z.almalki@psau.edu.sa (Z.S.A.); a.alfaifi@psau.edu.sa (A.A.A.); ah.alshehri@psau.edu.sa (A.M.A.); a.alahmari@psau.edu.sa (A.K.A.); m.balaha@psau.edu.sa (M.F.B.); 2Discipline of Clinical Pharmacy, School of Pharmaceutical Sciences, Universiti Sains Malaysia, George Town 11800, Penang, Malaysia; 3Infectious Diseases Department, Al-Kharj Military Industries Corporation Hospital, Al-Kharj 16417, Saudi Arabia; emad.m.elazab@gmail.com; 4Health Sciences Institute, Near East University, North Nicosia 99138, Cyprus; alamansour982@gmail.com; 5Clinical Pharmacy Department, College of Pharmacy, Umm Al-Qura University, Mecca 21955, Saudi Arabia; amhaseeb@uqu.edu.sa; 6Pharmacology Department, Faculty of Medicine, Tanta University, El-Gish Street, Tanta 31527, Egypt

**Keywords:** antimicrobial stewardship, surgical site infection, surgical antimicrobial prophylaxis

## Abstract

Perioperative antimicrobial prophylaxis is effective in reducing the rate of surgical site infections (SSIs); however, non-adherence to surgical antimicrobial prophylaxis protocols can lead to several negative outcomes. We performed a before-and-after intervention study with the aim of improving the process outcome, including adherence to guidelines. Another objective of this study was to investigate improvement in patient outcomes as a result of adherence to a surgical antimicrobial prophylaxis programme. The indicators of improved patient outcomes were a reduction in overall SSI rate and the decreased cost of antibiotics. SSI rate was calculated as a percentage by dividing the number of SSIs by the total number of surgeries and then multiplying the value obtained by 100%. The interventions implemented in the surgical antimicrobial prophylaxis programme included establishment of a guideline, educational sessions, and a monthly revision of prescriptions. Our findings show that implementation of the interventions resulted in reduced antibiotic consumption, a considerable decrease in the cost of prophylaxis, and a decrease in the incidence of SSIs.

## 1. Introduction

Surgical site infections (SSIs) are the most common healthcare-associated infections that occur among surgical patients. It is reported that SSIs lead to prolonged hospital stay, readmission to the hospital, and increased mortality and morbidity rates [1]. SSIs are defined as postoperative infections that occur within 30 days after undergoing a surgical procedure, or within 1 year after placement of a permanent implant [2]. It is indicated in a European Centre for Disease Prevention and Control report that the cumulative incidence of SSIs mainly depends on the type of surgical procedure; the highest rates have been reported for open colorectal and laparoscopic colorectal operations (10.1% and 6.4%, respectively), followed by open cholecystectomy (3.9%) and coronary artery bypass graft (2.6%) [2]. Studies on SSIs have largely focused on surgeries performed on the gastrointestinal tract, which can result in higher SSI rates compared to other procedures. Among the different types of gastrointestinal surgeries, colorectal surgery is the most concerning regarding the development of SSIs due to the presence of multiple microbes in the colon and rectum [3]. The number of bacteria and proportion of anaerobic bacteria progressively increase along the gastrointestinal tract; therefore, SSIs that develop after gastrointestinal surgeries must be treated based on the sites at which the procedures were performed [4].

Perioperative antimicrobial prophylaxis has been shown to be effective in reducing the rate of SSIs after surgical procedures [5]. However, inappropriate implementation of surgical antimicrobial prophylaxis, in terms of prolonged duration of the prophylactic treatment in addition to the use of broad-spectrum antibiotics, can result in the development of antibiotic resistance as well as high treatment costs [6]. Moreover, inappropriate timing of surgical antimicrobial prophylaxis can lead to a decrease in the efficacy of the prophylaxis [7]. Therefore, the quality of prophylaxis has been the subject of several audits [8,9,10,11,12] and interventional studies [13,14,15,16,17,18,19,20], and numerous national guidelines have been developed to support appropriate implementation of prophylaxis protocols [21,22,23,24].

Antimicrobial stewardship programmes (ASPs) are one of the most important aspects of multipronged interventions to decrease the incidence of SSIs as well as morbidity from antimicrobial-resistant SSIs [25]. The definition for antimicrobial stewardship is an organisational or healthcare system-wide approach to promote and monitor the judicious use of antimicrobials to preserve their future effectiveness [26].

Research on ASP implementation in Saudi Arabia and the effect of antimicrobial stewardship in surgical antibiotic prophylaxis in the region is scarce. To the best of our knowledge, this study is the first to assess the effect of ASP interventions on surgical antimicrobial prophylaxis in Al-Kharj city. 

The present study analysed data from various gastrointestinal surgery cases including colorectal surgeries, appendectomies, and bile duct/gall bladder and aimed to investigate the impact of interventions in a surgical antimicrobial prophylaxis programme on process outcome parameters (administration of prophylactic antibiotics) and patient outcome (reduction in SSI rate and cost of consumed antibiotics) at Al-Kharj Military Industries Corporation Hospital (Al-Kharj, Saudi Arabia). 

## 2. Materials and Methods

### 2.1. Setting

This study was a prospective interventional study performed at Al-Kharj Military Industries Corporation Hospital (Al-Kharj, Saudi Arabia). The management of this hospital is responsible for developing health care facilities as well as creating health awareness and providing medical care to members of the armed forces and their families.

### 2.2. Data Collection

During the pre- and post-intervention periods, all procedures meeting the inclusion criteria below were included in the analysis. Data were extracted from surgery files and electronic medical records. The data included type of surgery, date of surgery, patient’s age and gender, and the prescribed medications. 

The durations of the pre- and post-intervention phases of data collection depended on the number of procedures in each phase; as a result, there were variations in the data for the two phases. Activities for the pre-intervention phase started on 1 June 2019 and ended on 15 December 2019, whereas those for the post-intervention phase started on 16 December 2019 and ended on 30 July 2020. To obtain a balanced distribution of the selected procedures, the number of different gastrointestinal surgeries and the age and gender of patients were almost similar before and after the intervention.

### 2.3. Inclusion and Exclusion Criteria

The data on all colorectal surgeries, appendectomies, and bile duct/gall bladder surgeries that were performed during the study period were included in the analysis. Other types of surgeries and surgeries that were conducted before or after the study period were excluded from the analysis.

### 2.4. Interventions

A guideline on surgical antimicrobial prophylaxis was established, after which implementation of the interventions in the guideline was evaluated. The antibiotic use committee in the hospital announced the guideline to the clinical staff and after that several educational sessions were organized by physicians in the hospital and by an academic staff in the clinical pharmacy department at Prince Sattam Bin Abdulaziz university on antimicrobial stewardship and about the correct use of antibiotics as recommended in surgical antimicrobial prophylaxis guidelines. In addition, monthly reviews of prescribing patterns in the surgery department were conducted to assess improvement in the prescription of antibiotics, with respect to appropriateness of antibiotic selection, route of drug administration, timing of drug administration, antibiotic dose(s), and duration of prophylaxis and after that, feedback was sent to the surgeons.

### 2.5. Data Analysis

The main objectives of this study were to assess improvements in process and patient outcomes as a result of adherence to the guideline and implementation of the recommended protocols. The aim of the surgical antimicrobial prophylaxis programme was to reduce the overall SSI rate and decrease the cost of consumed antibiotics among patients. SSI rate was calculated as a percentage (%) by dividing the number of SSIs by the total number of surgeries and multiplying the value obtained by 100%.

Antimicrobial use was analysed quantitatively by calculating the defined daily dose (DDD) per 100 procedures. DDDs were obtained from the ATC/DDD Index 2003 of the World Health Organization Collaborating Centre for Drug Statistics Methodology [27]. The cost of consumed antibiotics was calculated by multiplying the DDD/100 surgeries by the cost of the antibiotics. Courses of antimicrobial drugs were reviewed for drug choice, administration route, dosage, and duration and timing of prophylaxis. If no antibiotic prescriptions were recorded, it was assumed that antibiotics had not been administered. All the results have been presented as numbers and percentages. The data were evaluated with the *F*-test, then were compared with either the two-sample Student’s *t*-test or the Mann–Whitney’s U test.

## 3. Results

### 3.1. Patients’ Characteristics

Table 1 shows the number and percentages of gastrointestinal surgeries that were included in the study with respect to the age ranges and gender of the patients. The number of surgeries before implementation of the guideline was 178; however, 184 surgeries were recorded after implementation of the guideline. More than half of the patients were female, and the majority of them were aged 20–49 years in both the pre- and post-intervention phases.

Table 2 shows the specific number and percentages of bile and gall bladder surgeries, colorectal surgeries, and appendectomies included in the study. The most performed surgeries in the pre-intervention phase were bile duct/gall bladder surgeries (46.07%), followed by appendectomies (39.33%) and colorectal surgeries (14.60%). The most performed gastrointestinal surgeries in the post-intervention phase were appendectomies (42.93%), followed by bile duct/gall bladder surgeries (41.31%) and colorectal surgeries (15.76%). 

### 3.2. Appropriateness of Antimicrobials Used for Prophylaxis 

Table 3 shows the frequency at which patients were administered antibiotics. The data show that for all of the included surgeries, patients had to be administered antibiotics. Additionally, antibiotics were not prescribed for the prophylaxis of infections in about 8.99% of surgeries before the intervention and in about 10.87% of surgeries after the intervention.

As shown in Table 4, implementation of the surgical antimicrobial prophylaxis programme resulted in improvement in correct antibiotic use. In addition, there were improvements in the prescription of appropriate antibiotic doses, timing of prophylaxis, choice of route of drug administration, and duration of prophylaxis. The appropriateness of drug selection increased from 51.23% before the intervention to 53.05% after the intervention. Appropriateness of drug dose also improved from 32.72% before the intervention to 53.66% after the intervention. The appropriateness of timing increased from 64.81% before implementation of the intervention to 74.39% after the intervention. Similarly, the appropriateness of route of drug administration and duration of prophylaxis improved from 66.67% to 76.83% and from 14.20% to 19.51%, respectively.

### 3.3. Reduction in SSI Rate

Table 5 shows the rate of SSIs in 2019 before implementation of the intervention, and in 2020 after implementation of the intervention. The total rate of SSIs in 2019 was 0.41%, whereas that in 2020 was 0.04% (this percentage was for the total rate of SSIs from all surgeries and not for gastrointestinal surgeries only).

### 3.4. Antimicrobial Use

According to the guideline, the first drug of choice for bile duct and gall bladder surgeries is cefazolin (the recommended antibiotic); however, if cefazolin is not available, cefuroxime or ceftriaxone (alternative agents) may be used. For colorectal surgeries and appendectomy, patients must be administered cefazolin and metronidazole (the recommended combination); however, if cefazolin is not available, cefuroxime or ceftriaxone and metronidazole (alternative agents) may be used. As shown in Table 6, in the post-intervention phase, the use of intravenous cefazolin increased (from 0 to 7.97 DDD/100 surgeries), whereas the use of intravenous cefuroxime decreased (from 13.76 to 0.27 DDD/100 surgeries). Moreover, the use of intravenous metronidazole increased (from 23.31 to 29.07 DDD/100 surgeries), whereas the total consumption of antibiotics reduced from 920.36 to 788.56 DDD/100 surgeries.

As shown in Table 7, the costs of cefuroxime, ceftriaxone, and metronidazole decreased after the intervention, whereas those of ciprofloxacin, Amoxicillin/Clavulanic acid, and cefazolin increased. The maximum estimation of reduction in cost between the pre- and post-intervention phases was 1783.81 Saudi Riyal (9513.92–7730.11) that is equal to 475.39 United States Dollar (*p*-value < 0.05). 

## 4. Discussion

Implementation of the surgical antimicrobial prophylaxis programme in the present study resulted in increased compliance of prescribers with the guideline recommendations. It also led to improvements in the correct use of antibiotics, appropriate dose prescription, timing and duration of prophylaxis, and choice of route of administration. However, there was still a problem with the duration of prophylaxis. In a previous study by Segala et al. [28], it was found that after the implementation of an antimicrobial stewardship intervention, the appropriateness of prophylaxis duration was still not optimal, which was the most reported reason for non-compliant prescribing in the post-interventional survey of their study. Elyasi et al. [29] have also reported that there was a significant reduction in incorrect use of antibiotics after the implementation of an evidence-based guideline for gastrointestinal surgeries in a teaching hospital. Additionally, implementation of the guideline resulted in a reduction in the duration of antibiotic prophylaxis and a decrease in the prescription of inappropriate antibiotic doses, although this was not significant [29].

Implementation of the prophylaxis programme in the present study resulted in decreases in SSI rate, overall antibiotic consumption, and the cost of antibiotics. Sarang et al. [25] have reported that implementing SSI prevention guidelines, including proper antimicrobial prophylaxis, in two tertiary-care hospitals in Mumbai resulted in a significant decrease in prophylaxis costs; however, the SSI rates did not differ from those obtained after following the same international standards. A study conducted by van Kasteren et al. [30] in a multi-site, interventional study in Dutch hospitals revealed that intervention implementation led to improved quality of surgical antimicrobial prophylaxis as well as a reduction in antibiotic use and cost; however, there was an insignificant reduction in SSIs. Similarly, optimising antibiotic prophylaxis resulted in an insignificant reduction in the overall SSI rate from 5.4% to 4.5% in another study [31]. Furthermore, Fujibayashi et al. [32] have suggested that revising all relevant clinical pathways in invasive therapies may be highly effective in reducing antibiotic consumption and shortening the duration of antibiotic administration. It was found in the study conducted by Fujibayashi et al. [32] that the incidence of SSIs was not significantly different before and after the revisions made.

A study performed in a private hospital in Saudi Arabia by Kilan et al. [33] showed that implementation of interventions that could improve surgical antibiotic prophylaxis resulted in improved antibiotic selection, dosing, and timing of prophylaxis from 47.3% to 82.2% in patients undergoing gastrointestinal surgery (*p* < 0.001) [33]. Additionally, implementation of the interventions led to a reduction in SSI rate, although this change was not statistically significant. 

### Limitations

The main limitation of the present study was that the intervention was not implemented in a control group, as that was considered unethical. Second limitation was that some of the resulting changes were not statistically significant because there was a limited number of gastrointestinal surgeries that were performed in the hospital.

## 5. Conclusions

Implementation of the surgical antimicrobial prophylaxis programme included the establishment of a guideline, educational sessions, and a monthly revision of prescriptions. Our findings show that implementation of the programme resulted in decreases in antibiotic consumption, antibiotic prophylaxis cost, and the incidence of SSIs.

## Figures and Tables

**Table 1 healthcare-10-00464-t001:** Patients who underwent gastrointestinal surgery before and after the intervention.

Variable	Category	Number of Patients in the Pre-Intervention Phase *n* (%)	Number of Patients in the Post-Intervention Phase *n* (%)
Gender	Male	85 (47.75)	86 (46.74)
	Female	93 (52.25)	98 (53.26)
Age (years)	<10	8 (4.49)	10 (5.43)
	10–19	29 (16.29)	29 (15.76)
	20–29	37 (20.79)	39 (21.20)
	30–39	48 (26.97)	44 (23.92)
	40–49	30 (16.86)	38 (20.65)
	50–59	19 (10.67)	15 (8.15)
	>59	7 (3.93)	9 (4.89)

**Table 2 healthcare-10-00464-t002:** Surgeries performed on the patients.

Type of Surgery	Number of Surgeries in the Pre-Intervention Phase *n* (%)	Number of Surgeries in the Post-Intervention Phase *n* (%)
Appendectomy	70 (39.33)	79 (42.93)
Bile duct/Gall bladder surgery	82 (46.07)	76 (41.31)
Colorectal surgery	26 (14.60)	29 (15.76)
Total	178	184

**Table 3 healthcare-10-00464-t003:** Frequency of antibiotic use for gastrointestinal surgeries during the pre- and post-intervention phases.

Surgery Type	Use of Antibiotic	Pre-Intervention Phase *n* (%)	Post-Intervention Phase *n* (%)
Appendectomy	Yes	68 (97.14)	71 (89.87)
	No	2 (2.86)	8 (10.13)
Bile duct/Gallbladder surgery	Yes	72 (87.80)	68 (89.47)
	No	10 (12.20)	8 (10.53)
Colorectal surgery	Yes	22 (84.62)	25 (86.21)
	No	4 (15.38)	4 (13.79)
All gastrointestinal surgeries	Yes	162 (91.01)	164 (89.13)
	No	16 (8.99)	20 (10.87)

**Table 4 healthcare-10-00464-t004:** Appropriateness of surgical antimicrobial prophylaxis in the pre- and post-intervention phases.

Surgery Type	Appropriateness of Surgical Antimicrobial Prophylaxis		Pre-Intervention Phase *n* (%)	Post-Intervention Phase *n* (%)	*p*-Value
Appendectomy	Selected drug(s)	Appropriate	22 (32.35)	29 (40.85)	0.049
		Inappropriate	46 (67.65)	42 (59.15)	
	Drug dose(s)	Appropriate	20 (29.41)	29 (40.85)	
		Inappropriate	48 (70.59)	42 (59.15)	
	Timing	Appropriate	44 (64.71)	55 (77.46)	
		Inappropriate	24 (35.29)	16 (22.54)	
	Route of drug administration	Appropriate	47 (69.12)	57 (80.28)	
		Inappropriate	21 (30.88)	14 (19.72)	
	Duration	Appropriate	2 (2.94)	5 (7.04)	
		Inappropriate	66 (97.06)	66 (92.96)	
Bile duct/Gall bladder surgery	Selected drug(s)	Appropriate	55 (76.39)	48 (70.59)	0.461
		Inappropriate	17 (23.61)	20 (29.41)	
	Drug dose(s)	Appropriate	24 (37.50)	38 (55.88)	
		Inappropriate	48 (62.50)	30 (44.12)	
	Timing	Appropriate	46 (63.89)	45 (66.18)	
		Inappropriate	26 (36.11)	23 (33.82)	
	Route of drug administration	Appropriate	46 (63.89)	46 (67.65)	
		Inappropriate	26 (36.11)	22 (32.35)	
	Duration	Appropriate	16 (22.22)	21 (30.88)	
		Inappropriate	56 (77.78)	47 (69.12)	
Colorectal surgery	Selected drug(s)	Appropriate	6 (27.27)	10 (40.00)	0.0153
		Inappropriate	16 (72.73)	15 (60.00)	
	Drug dose(s)	Appropriate	9 (40.91)	21 (84.00)	
		Inappropriate	13 (50.09)	4 (16.00)	
	Timing	Appropriate	15 (68.18)	22 (88.00)	
		Inappropriate	7 (31.82)	3 (12.00)	
	Route of drug administration	Appropriate	15 (68.18)	23 (92.00)	
		Inappropriate	7 (31.82)	2 (8.00)	
	Duration	Appropriate	5 (22.73)	6 (24.00)	
		Inappropriate	17 (77.27)	19 (76.00)	
		Inappropriate	139 (85.80)	132 (80.49)	

**Table 5 healthcare-10-00464-t005:** Number of SSIs recorded in 2019 and 2020.

Months	SSIs in 2019 (Pre-Intervention Period)		SSIs in 2020 (Post-Intervention Period)	
	Number of SSIs/Total Number of Surgeries	SSI Rate	Number of SSIs/Total Number of Surgeries	SSI Rate
January–March	3/635	0.47%	0/666	0.00%
April–June	2/639	0.31%	0/469	0.00%
July–September	3/610	0.49%	1/550	0.18%
October–December	3/811	0.37%	0/706	0.00%
Total	11/2695	0.41%	1/2391	0.04%

**Table 6 healthcare-10-00464-t006:** Number of patients who were administered different antibiotics before and after the intervention.

Antibiotic	Route of Drug Administration	DDD of Antibiotic before the Intervention	DDD/100 Surgeries before the Intervention	DDD of Antibiotic after the Intervention	DDD/100 Surgeries after the Intervention
Ceftriaxone	Intravenous	1.50	0.84	1	0.54
Cefuroxime	Intravenous	24.5	13.76	0.50	0.27
	Oral	1414	794.38	994	540.22
Metronidazole	Intravenous	41.50	23.31	53.50	29.07
	Oral	71.4	40.11	63	34.24
Ciprofloxacin	Intravenous	2.25	1.26	0.25	0.14
	Oral	1	0.56	47.50	25.82
Amoxicillin/Clavulanic acid	Oral	62.12	34.90	232.53	126.38
Cefazolin	Intravenous	0	0	14.67	7.97
Doxycycline	Oral	0	0	42	22.83
Trimethoprim/Sulfamethoxazole	Oral	0	0	2	1.09
Cefprozil	Oral	20	11.24	0	0
Total		1638.27	920.36	1450.95	788.56

**Table 7 healthcare-10-00464-t007:** Antibiotic cost before and after the intervention.

Antibiotics	Route of Drug Administration	DDD/100 Surgeries before the Intervention	Antibiotic Cost in the Pre-Intervention Phase	DDD/100 Surgeries after the Intervention	Antibiotic Cost in the Post-Intervention Phase
Ceftriaxone	Intravenous	0.84	77.28	0.54	49.68
Cefuroxime	Intravenous	13.76	598.42	0.27	11.74
	Oral	794.38	6156.45	540.22	4186.71
Metronidazole	Intravenous	23.31	736.60	29.07	918.61
	Oral	40.11	768.11	34.24	655.70
Ciprofloxacin	Intravenous	1.26	199.95	0.14	22.22
	Oral	0.56	7.83	25.82	361.22
Amoxicillin/Clavulanic acid	Oral	34.90	244.30	126.38	884.66
Cefazolin	Intravenous	0	0	7.97	89.26
Doxycycline	Oral	0	0	22.83	531.94
Trimethoprim/Sulfamethoxazole	Oral	0	0	1.09	18.37
Cefprozil	Oral	11.24	724.98	0	0
Total		920.36	9513.92	788.56	7730.11

## Data Availability

Not applicable.

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
