# Peer review of "Implementing an Antimicrobial Stewardship Programme to Improve Adherence to a Perioperative Prophylaxis Guideline"

_healthcare, 2022, doi:10.3390/healthcare10030464_

Round 1
Reviewer 1 Report
Antibiotic Stewardship implies a set of key principles to guide efforts to improve antibiotic use, desiring to facilitate and advance patient safety and improve outcomes. International Guidelines recognize that there is no “one size fits all” approach to optimize antibiotic use for all medical settings. The complexity of medical decision-making surrounding antibiotic use and the variability in facility size and types of care require flexible programs and activities. Local guidelines must be implemented, and their recommendations must stand o a large base of local antimicrobial resistance knowledge. Different patterns of microbial prevalence must be transformed in particular guidelines. Research involving results derived from stewardship measures is still scarce and limited. The article pioneers in assessing Antibiotic Stewardship effects as an intervention on surgical antimicrobial prophylaxis in Al-Kharj city, Saudi Arabia.
I have a few points to raise for the authors.
Point 1 I suggest to precisely tell in the introduction section that this article analyzes data from general surgery cases, aka colorectal surgeries, appendectomies, and bile duct/gall bladder.
Point 2 Lines 144-145 from the Discussions section declare that correct implementation of the surgical antimicrobial prophylaxis resulted in improved antibiotic use. Still, a thorough read of tables 3, 4, and 6 demonstrates only a minor percentage increment. Perhaps inserting a new subsection – Limitations would be a good idea, and it should detail why the interventions were implemented in a limited quota. Thus, the resulting changes were not statistically significant.
Point 3 Line 178 Convert Saudy Rial in an international currency for better understanding the costs.
Point 4 The results presented do not sustain the conclusions drawn.
Author Response
Point 1 I suggest to precisely tell in the introduction section that this article analyzes data from general surgery cases, aka colorectal surgeries, appendectomies, and bile duct/gall bladder.
Yes I add the sentence in the introduction part
Point 2 Lines 144-145 from the Discussions section declare that correct implementation of the surgical antimicrobial prophylaxis resulted in improved antibiotic use. Still, a thorough read of tables 3, 4, and 6 demonstrates only a minor percentage increment. Perhaps inserting a new subsection – Limitations would be a good idea, and it should detail why the interventions were implemented in a limited quota. Thus, the resulting changes were not statistically significant.
I add this point in the limitation
Point 3 Line 178 Convert Saudy Rial in an international currency for better understanding the costs.
I convert the currency to Dollar
Point 4 The results presented do not sustain the conclusions drawn.
In the conclusion “Our findings show that implementation of the programme resulted in decreases in antibiotic consumption, antibiotic prophylaxis cost, and the incidence of SSIs” as mentions in the results but I add in the limitations that the number of surgeries is low and also the rate of SSIs is low so the results were not statistically significant.
Reviewer 2 Report
Thanks for this study. Antibiotic prophylaxis for abdo surgery is very common- therefore, research on interventions that could improve the quality of prophylaxis is pertinent. In my opinion, there are some opportunities to increase the scientific validness of your manuscript:
1) Could you provide more details about the intervention- as only the establishment of a new guideline is unlikely to do the trick. How was this guideline announced to the clinical staff? Were there any presentations in the beginning? Did prescribers get any feedback?
2) In order to understand the significance of the different changes- appropriate statistical tests should be conducted. Please consult a statistician for this- and report accordingly.
3) I would recommend English language support to increase the readability of the manuscript.
Author Response
1) Could you provide more details about the intervention- as only the establishment of a new guideline is unlikely to do the trick. How was this guideline announced to the clinical staff? Were there any presentations in the beginning? Did prescribers get any feedback?
I added more details about the guideline in red color.
2) In order to understand the significance of the different changes- appropriate statistical tests should be
The data were evaluated with the F- test, then compared with either the two-sample Student’s t-test or the Mann-Whitney’s U test to find the significance of the changes in the appropriateness and cost for each type of the surgeries
3) I would recommend English language support to increase the readability of the manuscript.
I sent the paper to editing institution before submission
Round 2
Reviewer 1 Report
The authors improved the overall quality of their article. And although, as they stated in the Limitations sub-section, the number of patients is rather small, the topic is of great importance, and data from that particular geographical area must be published. I still have a few observations to address them.
Point 1 The newly introduced material from lines 105 - 110 should be rephrased, as the actual form does not make enough sense.
Point 2 Line 190 - p-value is spelled incorrectly.
Point 3 Please revisit the English language from lines 230-234.
Author Response
Point 1 The newly introduced material from lines 105 - 110 should be rephrased, as the actual form does not make enough sense.
I rephrase the complete paragraph
Point 2 Line 190 - p-value is spelled incorrectly.
I corrected it
Point 3 Please revisit the English language from lines 230-234.
I rephrase these lines
Reviewer 2 Report
- Page 3 line 105: would suggest to put 'The' before antibiotic use committee.
- Page 6 last paragraph: I would suggest to add the differences in cost to the p-values.
Author Response
- Page 3 line 105: would suggest to put 'The' before antibiotic use committee.
I add “The”
- Page 6 last paragraph: I would suggest to add the differences in cost to the p-values.
- I add the differences of the total cost.